# Role of PET/CT in Oropharyngeal Cancers

**DOI:** 10.3390/cancers15092651

**Published:** 2023-05-08

**Authors:** Emily W. Avery, Kavita Joshi, Saral Mehra, Amit Mahajan

**Affiliations:** 1Department of Radiology and Biomedical Imaging, Yale University School of Medicine, New Haven, CT 06520, USA; 2Department of Otolaryngology, Yale University School of Medicine, New Haven, CT 06520, USA

**Keywords:** oropharyngeal cancer, oropharyngeal squamous cell carcinoma, PET/CT

## Abstract

**Simple Summary:**

This article reviews the role of positron emission tomography/computed tomography (PET/CT) in the diagnosis and management of oropharyngeal cancers. Oropharyngeal cancer epidemiology, staging, treatment, response assessment, and disease monitoring are discussed. With regard to staging, response assessment, and monitoring, PET/CT has emerged as the modality of choice in recent decades. This is because PET/CT allows for the combination of precise anatomical information with metabolic activity of the tumor and surrounding structures. In oropharyngeal cancer, PET/CT is currently used to facilitate TNM (tumor, node, metastases) staging, assess tumor response after initial treatment, and monitor long-term for disease recurrence. Innovations in the field continue to improve the utility of PET/CT, and, most recently, applications of artificial intelligence and the development of novel therapeutic tracers have indicated promising new frontiers. Combined with continued advances in surgical, medical, and radiation treatment options, PET/CT is well-positioned to facilitate continued improvement of oropharyngeal cancer care.

**Abstract:**

Oropharyngeal squamous cell carcinoma (OPSCC) comprises cancers of the tonsils, tongue base, soft palate, and uvula. The staging of oropharyngeal cancers varies depending upon the presence or absence of human papillomavirus (HPV)-directed pathogenesis. The incidence of HPV-associated oropharyngeal cancer (HPV + OPSCC) is expected to continue to rise over the coming decades. PET/CT is a useful modality for the diagnosis, staging, and follow up of patients with oropharyngeal cancers undergoing treatment and surveillance.

## 1. Introduction

Head and neck cancers constitute a significant proportion of malignancies globally. In the United States alone, over 140 cases of head and neck cancers are diagnosed each day [1]. While advances in medical and surgical treatment options continue to improve survival and quality of life for head and neck cancer patients, treatment decision-making and prognostication is highly dependent on appropriate initial staging, re-staging, and disease monitoring.

In recent decades, 18F-fluorodeoxyglucose (FDG) positron emission tomography/computed tomography (PET/CT) has emerged as the mainstay for this purpose. This combination modality draws on two primary techniques: CT, which provides precise anatomic information about tumor boundaries and soft tissue characteristics, and PET, which measures metabolic activity of the tumor and surrounding structures. Together, PET/CT enables assessment of primary tumor location, malignant lymph node transformation, and possible distant metastasis or second primary.

In this review, we discuss the history and implementation of PET/CT in oropharyngeal cancer staging, diagnosis, and management, as well as recent and upcoming advancements in artificial intelligence and machine learning techniques to improve the utility and efficiency of this modality.

## 2. History of PET/CT Development

David Townsend, Ronald Nutt, and colleagues at the University of Geneva first proposed PET/CT as a dual-modality imaging technique in 1991. Inspired by low-cost, rotating PET scanners of the era, the PET/CT design utilized two rotating blocks of bismuth germanate (BGO) detectors [2]. The gaps between the BGO banks left room to mount the components of a CT scanner, to provide anatomical information in a format already familiar to physicians. With refinement, to accommodate the density of X-ray components for a spiral CT configuration, the first operational PET/CT scanner was constructed by CTI PET Systems and installed at the University of Pittsburgh in 1998.

The utility of PET/CT for oncology was demonstrated immediately. Over 300 cancer patients were scanned on the prototype machine, including many head and neck cancer patients, and numerous publications highlighted the advantage of PET/CT in this cohort [3,4,5]. GE Healthcare, with Siemens Medical Solutions and Philips Medical Systems quickly following suit, announcing the first commercial PET/CT scanner in 2001. These commercial scanners utilized the highest-resolution CT technology available at the time (four-slice) and have continued to advance in terms of spatial resolution, acquisition time, temporal resolution, and correction for respiratory movement. Since 2002, PET/CT has been one of the fastest-growing medical imaging modalities and has been transformative in oncologic care.

## 3. Oropharyngeal Cancer Epidemiology and Staging

Approximately 90% of head and neck cancers in the United States are squamous cell carcinomas (HNSCC), with the remaining 10% consisting mostly of lymphomas, sarcomas, melanomas, and salivary gland neoplasms [6]. HNSCC has been declining in prevalence for the past 40 years, largely due to decreasing rates of smoking [7]. Simultaneously, however, the incidence of oropharyngeal squamous cell carcinoma (OPSCC) has increased. In Western countries, the increase in OPSCC has been due to an increase in HPV-associated malignancy; to which over 70% of OPSCC are now attributable [7]. The disease seems to most affect white men although incidence in white women appears to be increasing as well, albeit at a lower rate. The incidence of human papillomavirus-associated oropharyngeal cancer (HPV + OPSCC) is expected to continue to rise over the coming decades until the benefits of gender-neutral prophylactic HPV vaccination become manifest. A recent study demonstrated lower incidence in young white men (HPV vaccination era) compared with the pre-vaccination era, suggesting the potential benefit effects of HPV vaccination [6]. Mortality for HPV-positive OPSCC is lower than that of HPV-negative disease, with 5-year survival of 80–90% even with lymph node involvement and 60% with N3 or M1 disease [8,9,10]. HPV-positive patients had a lower risk of cumulative incidence of all-cause mortality (10.4% vs. 33.3%) and head and neck cancer-specific mortality (4.8% vs. 16.2%) [11]. HPV-negative OPSCC continues to have a relatively poor prognosis, with a 5-year survival of around 67%. Current challenges in HPV-positive OPSCC are in de-intensifying treatment, surveillance, and treatment of recurrent or persistent disease. Given this disease’s severity and treatment complexity, it is critical to optimize early detection and staging, treatment decision-making, and disease monitoring and surveillance. PET/CT has grown to play an important role in this regard.

## 4. Oropharyngeal Anatomy

OPSCC comprises cancers of the tonsils, base of tongue, soft palate, and uvula. Anatomically, the oropharynx is defined by its boundaries including the circumvallate papilla (anterior), vallecula (inferior), hyoid bone, and soft palate (superior). The oropharynx is further defined by four specific subsites: (1) the base of the tongue, (2) the palatine tonsils, (3) the ventral soft palate, and (4) the posterior and lateral pharyngeal walls (Figure 1: CT anatomy of the oropharynx). While these regions are composed of mucosal, lymphoid, salivary, and fascial tissue, squamous cell carcinoma arising from mucosal surfaces is the dominant tumor pathology [12].

## 5. Imaging of Oropharyngeal Cancers

HPV-negative OP cancers can arise from any subsite of the oropharynx while HPV-positive cancers are most common in the tonsillar and base of tongue subsites (Figure 2). Nodal involvement in HPV-related disease often leads to cystic or necrotic large lymph nodal masses. The oropharyngeal primary lesions are often small and constitute a large proportion of head and neck cancers with unknown primary. Addition of PET/CT improves detection or primary cancer by up to 54% [13]. PET/CT also has the added ability to detect unexpected contralateral nodal disease, synchronous malignancies, and distant lymph node involvement.

## 6. Staging of Oropharyngeal Cancers

While clinical factors such as age, sex, comorbidities, race, and tumor biology all contribute to prognosis, anatomic staging is the driver for prognosis-based treatment decisions and should be performed at the time of patient presentation. Anatomy-based staging is the basis for the American Joint Committee on Cancer (AJCC) staging, now in its eighth edition [14]. This is based on the description of the size and spread of primary tumor (T), nodal involvement (N) and distant metastases (M), TNM staging, which dictates the overall final tumor staging. Imaging plays a major role in accurate staging of head and neck tumors. Uncertain cases should be assigned to the lower-severity TNM category, and co-occurring primary tumors must be staged separately.

*Tumor, “T”*: indicates the size and spread of the primary mass lesion. For example, T1 (least advanced) indicates a lesion less than 2 cm, while T4 indicates a lesion which invades adjacent structures. A distinction is made between moderately advanced local disease, T4a, and very advanced local disease, T4b, especially for HPV-negative tumors, wherein T4b tumors extend into the skull base, lateral nasopharynx, lateral pterygoid muscle, pterygoid plates, or internal carotid artery. Accurate T stage assessment of tumor borders and extension is critical for surgical planning to achieve negative margins. MRI is sometimes preferred for this purpose, due to its superiority in delineating soft tissue anatomy and tumor boundaries, particularly in cases with excessive dental artifact. However, most work indicates the similar accuracy of CT and MRI for tumor delineation during T stage assessment [15]. Direct visualization via endoscopic or fiberoptic exam, prior to treatment, should also be performed when clinically indicated to assess accuracy of tumor size and extent.

*Node, “N”*: indicates the extent of regional lymph node spread. For example, N0 (least advanced) indicates no nodal spread, while N3 (most advanced) indicates metastasis to a regional node measuring more than 6 cm. Extranodal extension of tumor (ENE+) surrounding a lymph node is also assessed, both clinically and radiologically. On imaging, this may be seen as irregular node margins or conglomeration of nodes. When present, ENE indicates a worse prognosis. PET/CT can be helpful in determining nodal spread and is often used for this aspect of staging. As described below, nodal staging for HPV-positive OPC differs from HPV-negative cases and is determined by both number and size of nodes, rather than size alone and the presence or absence of ENE.

*Metastases, “M”*: indicates the absence (M0) or presence (M1) of distant metastasis in binary form. PET/CT may be useful for assessment of metastases in many cases, though the benefit of PET/CT over chest CT alone is lower in patients with a low probability of metastatic disease.

The final accurate determination of TNM staging requires pathological confirmation of malignancy and assessment of surgical or biopsy specimens for HPV-associated markers such as p16 expression and HPV DNA detection.

For HPV-positive OPC cases, staging should be altered such that: (1) T stage classifications of Tis and T4b are not used, (2) N stage classification is based on number of nodes, size and laterality, (3) overall stage IV classification is only given to cases with distant metastasis (M1) [10]. Complete staging guidelines for HPV-positive and HPV-negative OPC can be referenced in the AJCC Cancer Staging Manual [14].

## 7. Treatment

For localized T1–T2 disease, without nodal involvement, radiation therapy (RT) or surgery are both treatment options. Surgery involves resection of primary and ipsilateral or bilateral neck dissection. For N1 disease, either nodal dissection alone, radiation alone, nodal dissection with radiation +/− chemotherapy, or radiation + chemotherapy can all be suitable treatment options depending on T stage, positive node location, and additional pathologic findings such as extranodal extension and margin status. For HPV-positive disease, treatment de-intensification approaches have been shown to be effective in appropriately risk-stratified patients [16]. For advanced-stage disease, including T3–T4 tumors, high nodal burden, and/or those with clinical extranodal extension; concurrent systemic therapy and RT are usually considered as initial treatment, especially for HPV-positive disease. While various treatment de-escalation approaches for HPV-positive disease such as substantially reduced radiation dosages are being actively studied [16,17], HPV-negative disease continues to require intense treatment due to the significantly worse prognosis.

For HPV-positive and HPV-negative patients alike, PET/CT has more recently optimized RT by enabling dose-painting. This methodology uses local PET voxel intensities to guide radiation dose thresholds, wherein higher-intensity voxels require higher radiation and vice versa [18]. Dose-painting has demonstrated similar locoregional disease control to standard RT without increasing acute toxicity [19] and is being investigated for potential improvement in clinical outcomes [20].

## 8. Role of PET/CT

PET/CT can be used for initial staging or follow up of patients with oropharyngeal cancers.

### 8.1. Initial Staging

When used for initial staging, PET/CT was found to have a specificity and sensitivity of 95.2 and 80%, resulting in a change in TNM staging and an alteration in treatment in 16% of cases [21]. When compared to pathology, poor sensitivity of PET/CT to detect micrometastases has been described. Although the role of PET/CT in initial staging is unclear [22], the ACRIN 6685 study demonstrated a high negative predictive value of PET/CT for the clinically N0 neck (87%) in T2–T4 HNSCC. The optimal cutoff maximum standardized uptake value was determined to be 1.8, with an NPV of 94%. The surgical treatment plans based on PET/CT findings were changed in 22% of cases [23]. Complementary roles of PET/CT and conventional imaging have also been reported [24].

The National Comprehensive Cancer Network (NCCN) [25] suggests using FDG PET/CT for:Patients with multistation or lower neck nodal involvement or high-grade tumor histology.To determine the surgical approach to the contralateral neck in tumors approaching the midline, to identify involved lymph nodes.For patients with locoregionally advanced cancer (e.g., T3–T4 primary or ≥N1 nodal staging), FDG PET/CT is preferred to evaluate for distant disease and thoracic metastases. In lieu of FDG PET/CT, CT of the chest with contrast is suggested to assess for presence of lung metastases and mediastinal nodal involvement.Detection of occult primary lesions in patients presenting with large nodal metastatic masses without identifiable primary cancer on conventional imaging (CT/MRI) before EUA, biopsies, and tonsillectomy.

### 8.2. Response Assessment

Assessment of treatment response requires clinical response based on symptoms and physical examination, post-treatment imaging findings, and newly emerging evidence for laboratory-based response indices for HPV-associated OPSCC [26]. Post-treatment imaging is well addressed by PET/CT, where early detection of residual disease can enable time-sensitive pursuit of salvage therapy.

One of the earliest prominent studies in this area by McCollum et al. [27] in 2004 indicated 100% sensitivity and 65% specificity for PET/CT detection of persistent disease at the primary site after induction chemotherapy. After chemoradiation completion, this work indicated a sensitivity of 67% and specificity of 53% for detection of persistent disease in cervical lymph nodes, with inflammatory changes thought to contribute to false-positive interpretations. Subsequent work has further quantified PET/CT performance for post-treatment response, with a 2011 meta-analysis by Gupta et al. [28] revealing a negative predictive value range of 93–96% for the primary site or neck lymph nodes. However, a positive predictive value range of 53–65% was also described by this study, again pointing to a high rate of false positives. This is usually attributed to inflammatory response to treatment. The consensus within the field remains that PET/CT adds value to contrast-enhanced CT alone for assessment OPSCC treatment response. PET/CT-guided surveillance has also demonstrated no significant difference in overall survival compared to planned neck dissection in patients with stage N2 or N3 disease [29], indicating the cost-effective advantage of PET/CT-guided surveillance and its potential to minimize surgical intervention.

To improve PET/CT performance for response assessment and disease monitoring, semi-quantitative evaluation methods have been proposed. Namely, standardized uptake values (SUVs) have been utilized to quantitatively characterize imaging findings. Kim et al. demonstrated decreased survival in 221 patients with OPSCC on pretreatment PET with advanced age >60 years; advanced tumor stage; and high tumor SUV_max_, SUV_peak_, metabolic tumor volume (MTV) and total lesion glycolysis (TLG) [30]. Parameters for 18F-FDG-PET/CT are lower in HPV-positive than in HPV-negative patients [31]. In one study, the maximum SUV in HPV-positive tumors was 3.9 units lower than in HPV-negative tumors. Nodal PET/CT parameters, however, were found to be higher in the HPV-positive group [32]. However, the use of SUV_max_ is felt to be a surrogate for advanced stage and has not demonstrated accuracy in predicting patient outcome [33]. The exact metabolic parameters remain unclear [34]. MTV and TLG are felt to be superior to SUV_max_ as prognostic biomarkers to predict outcome in patients with head and neck cancer [35]. Thus, qualitative interpretation criteria continue to be used [36]. Such qualitative criteria include the size and location of any abnormal uptake, where a decrease in size or intensity indicates treatment response and an increase indicates disease progression. Other criteria include the presence or absence of new abnormal uptake, as well as the overall uptake distribution within the region. As noted, inflammatory changes in the treatment area may contribute to false-positive follow-up interpretations (Figure 3a and Figure 4). Conversely, post-treatment changes such as tissue necrosis may lead to false-negative results [28] (Figure 3b). Potential confounding tissue changes must thus be carefully considered during interpretation of PET/CT follow-up, and correlation with clinical and lab-based indices of response is recommended. Nodal detection may be improved by using a nodal SUV_max_ and nodal SUV_max_/liver SUV_max_ in the pre-operative detection of metastatic nodes. The latter is considered superior to visual inspection as it may correct for inter-scanner variability [37].

The timing of post-treatment imaging is important: imaging too early may show incomplete treatment effect, but imaging too late risks further spread of treatment-resistant disease. Prominent previous work has indicated that follow-up PET/CT is most appropriately timed at 12 weeks after chemoradiation therapy to assess treatment response [38,39]. However, other studies have suggested time windows ranging from 7 to 14 weeks [40,41], and no clear consensus for follow-up PET/CT timing has been established. Persistent lymph node disease at the time of follow-up imaging is most commonly managed by neck dissection [25].

## 9. Disease Monitoring

The increasing prevalence of HPV-positive disease, improved disease detection, and advances in treatment options have all contributed to an increasing number of head and neck cancer survivors. In spite of the favorable prognosis of HPV + OPSCC, 10–25% of patients will develop disease recurrence within 2–5 years after initial diagnosis and treatment [8]. The NCCN provides guidelines for follow-up and surveillance of patients after initial successful treatment. While the NCCN notes that survival has not been linked with intensity of clinical follow-up [42,43], this organization continues to recommend regular clinical follow-up with a decrease in frequency over time [38].

NCCN recommends that first post-treatment imaging be performed within 6 months, without reference to imaging modality, optimally at 12 weeks, for T3 or T4, or N2-N3 disease of the oropharynx, to serve as an important baseline for treatment decision-making, prognostication, and cost-effective planning. Thereafter, subsequent imaging surveillance for head and neck cancer survivors is not recommended by the NCCN. However, a 2015 survey study found that 79% of head and neck cancer surgeons and radiation oncologists perform surveillance PET/CT scans for asymptomatic survivors, with 39% doing so for more than half of their patients [44].

Multiple clinical studies have examined the utility of long-term surveillance PET/CT in asymptomatic patients [18]. In 2013, Dunsky et al. found 24 patients out of 120 to have positive lesions (20%), with 6 having locoregional recurrence and 2 being taken for additional surgical treatment [45]. Similarly, in 2012, Beswick et al. noted recurrence in 73 out of 388 asymptomatic patients (19%), with 67 having locoregional recurrence [46]. With regard to survival, however, recent work has shown no advantage when disease recurrence is detected by PET/CT vs. clinical exam [40], or when distant metastasis is detected by PET/CT vs. other imaging modalities [47] in asymptomatic patients on long-term surveillance imaging follow-up.

Long-term surveillance imaging may provide psychological benefit to the patient; however, this must be balanced by the harm caused by over-imaging, unnecessary radiation exposure, follow-up procedures, and financial considerations. While routine long-term PET/CT is not an official recommendation for head and neck cancer survivors, there is a group of patients for whom it may be beneficial. Identification of this particular subset has yet to be defined, and this remains an important future direction of research.

## 10. Recent Advances, Conclusions, and Future Directions

Innovation in image acquisition and interpretation continues to advance the utility of PET/CT and its applications to head and neck cancer management. In particular, the implementation of artificial intelligence (AI) tools has pushed the field forward in recent decades. These could include tools to assess the primary tumor or the involved lymph nodes [48,49,50].

Several such tools take advantage of a technique known as radiomics, a quantitative approach wherein a large number of features are extracted from medical images and used to detect clinically relevant information that is invisible to the naked eye [51]. For example, a 2020 study by Haider et al. demonstrated a PET/CT radiomics signature for HPV association of primary tumors and metastatic cervical nodes in OPSCC [52]. Though PET/CT radiomics cannot yet replace tissue sampling, imaging tools such as this may pave the way towards minimizing invasive procedures during initial diagnosis. Other prominent studies have successfully utilized PET/CT radiomics to predict patient outcome [53,54] and improve tumor delination [55] and automatic tumor segmentation [56].

Other artificial intelligence methods have demonstrated promise in improving the capabilities of PET/CT data. Namely, deep learning techniques, which utilize artificial neural networks to extract progressively higher levels of features from their input data, have proven particularly useful. For example, while delineation of tumor volumes is both imperative and time-consuming, researchers such as Huang et al. in 2018 [57] and Moe et al. in 2021 [58] have utilized PET/CT-based convolutional neural networks (CNNs) to perform high-quality and precise automated tumor delineations. PET/CT CNNs have also demonstrated utility in improving patient prognostication [59,60] and predicting individual response to chemotherapy [61]. The ability of AI tools to improve both the speed and quality of diagnostic interpretations maximizes the utility of PET/CT for both patients and providers and will continue to do so as advances in AI techniques continue to be made.

Other upcoming advancements in PET/CT technology focus on improved image acquisition. New tracers such as 68Ga inhibitors of fibroblast activation protein have recently demonstrated precise detection and improved tumor delineation, identifying cancerous tissue that is not visible on CT or MRI and clearly differentiating cancer from healthy tissue [62]. These tracers may have a therapeutic benefit as well. Immunotherapy-linked PET/CT also holds promise in optimizing treatment timing [63]. Simultaneously, scanners that are faster, more precise, and easier to use continue to be developed, which will also contribute to improved diagnosis and treatment of head and neck cancer in the future.

In light of these recent and upcoming advances in PET/CT technology and interpretation, it is clear that this modality will continue to play a vital role in the diagnosis and management of head and neck cancers. The rapidly developing combination of PET/CT with AI methods is particularly promising and has the potential to significantly improve patient outcomes in the future. PET/CT has reshaped the field of head and neck cancer management and is well-positioned to continue to do so in years to come.

## Figures and Tables

**Figure 1 cancers-15-02651-f001:**
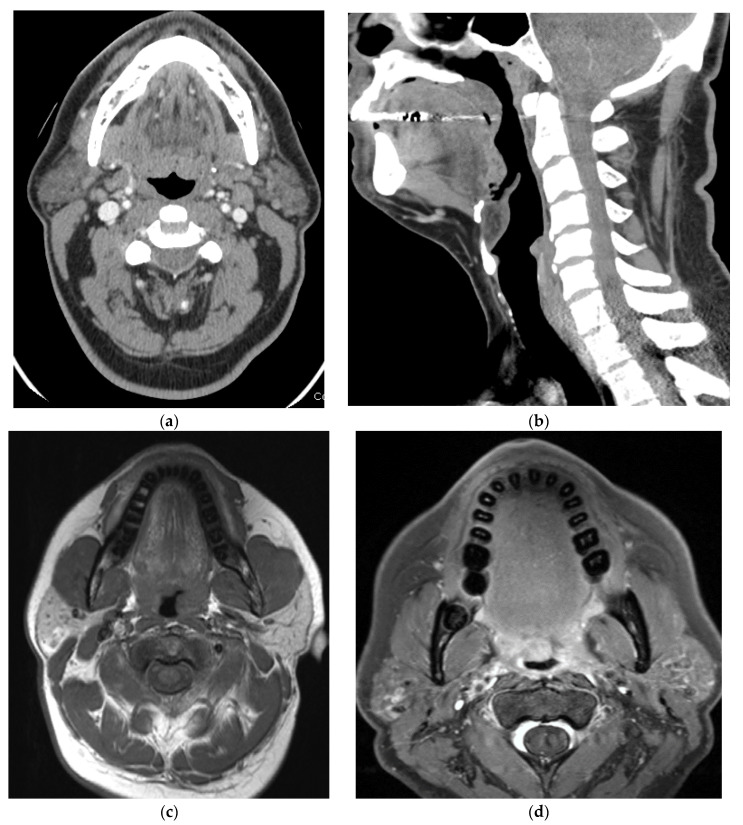
CT anatomy of the oropharynx on CT (**a**,**b**), MRI—Axial T1 weighted (**c**) and Fat-saturated Post-contrast axial (**d**) and PET/CT (**e**) showing physiologic FDG uptake in both tonsils.

**Figure 2 cancers-15-02651-f002:**
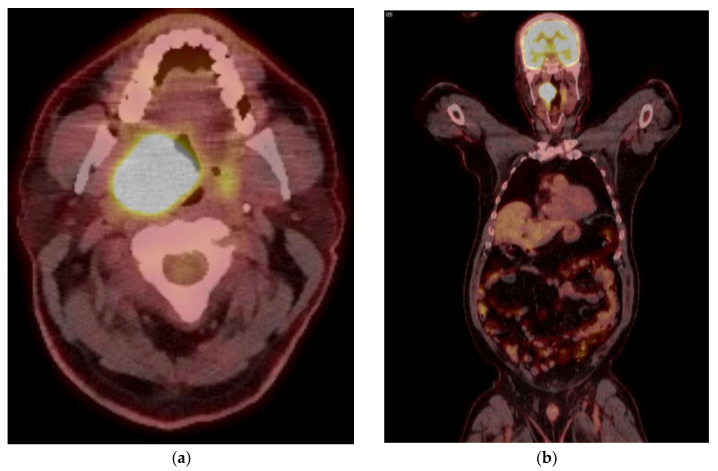
PET/CT images of head and neck cancer at time of diagnosis of a 55-year-old male with p16 positive oropharyngeal cancer: (**a**) Primary tumor showing a SUV_max_ of 30.8, Left tonsil—5.8, Blood pool—2.5, Liver—3.6, Bilateral Level 2 lymph nodes—2.6, with a lack of distant metastases (**b**). Concurrent contrast enhanced CT (**c**,**d**) shows a large tonsillar mass extending superiorly towards the soft palate. Follow up PET CT after 12 weeks showing response to chemoradiation therapy (**e**), with minimal residual uptake.

**Figure 3 cancers-15-02651-f003:**
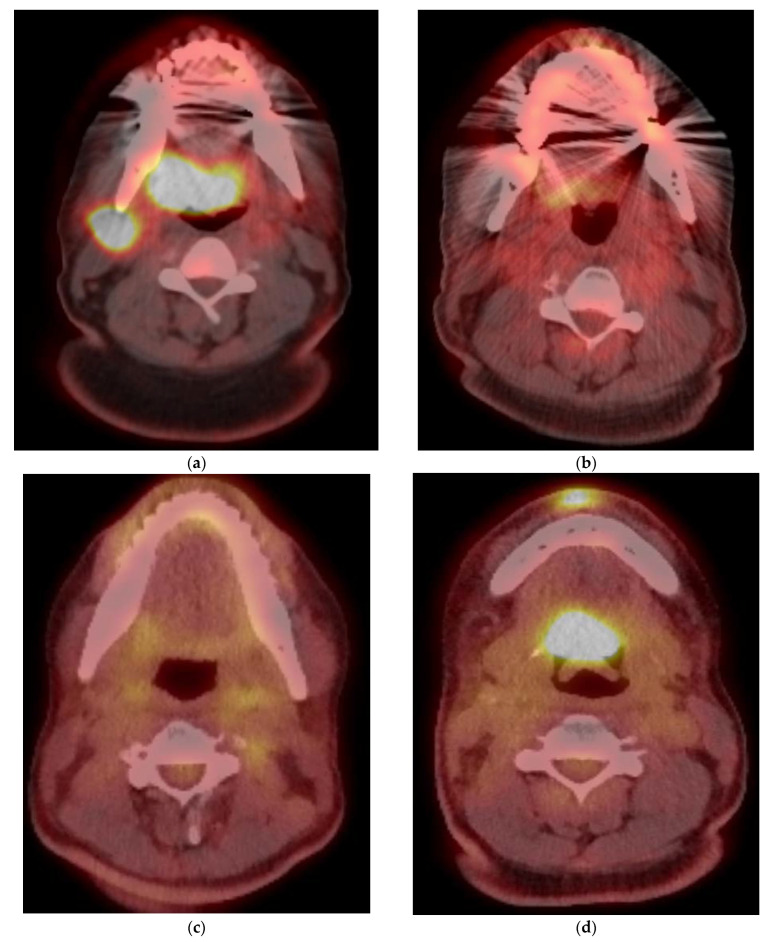
Pre- (**a**) and post- (**b**,**c**) treatment PET/CT image showing initial incomplete (**b**) and delayed complete (**c**) response to treatment, with subsequent recurrence on follow up PET/CT (**d**). This is confirmed on follow up anatomical imaging with contrast enhanced CT (**e**) and MRI (**f**).

**Figure 4 cancers-15-02651-f004:**
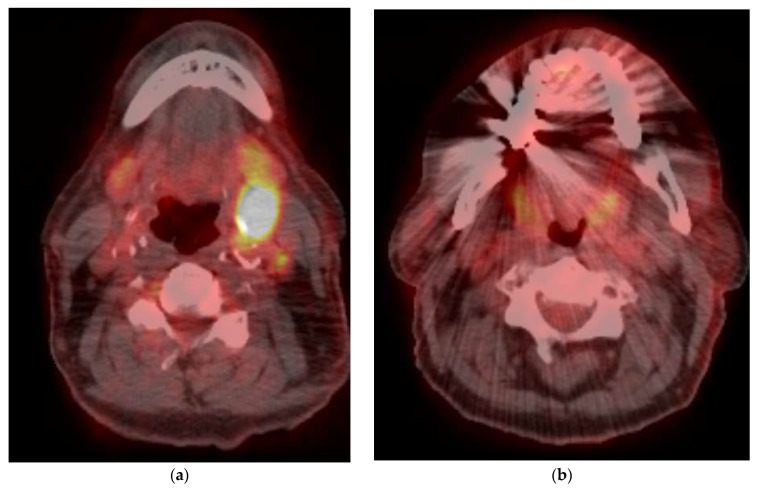
A 66-year-old male presenting with neck mass, found to have a nodal mass at level II (**a**) which is hypermetabolic on PET/CT (**b**), with a SUV_max_ of 20. No definite mass or FDG uptake was noted in the tonsillar fossa (SUV_max_-4). Follow up scan 12 weeks (**c**,**d**) after completion of Chemo-RT showed resolution of FDG uptake in the left neck lymph node with uptake in soft palate and base tongue, SUV_max_-8, without associated mass (False positive). Another follow up scan 6 months after completion of treatment (**e**), reveals resolution of oropharyngeal activity with new activity in the right neck, with a SUV_max_ of 3.9. Concurrent contrast enhanced CT (**f**) demonstrates this to be IJ thrombosis related to chest port.

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
