# Peer review of "Role of PET/CT in Oropharyngeal Cancers"

_cancers, 2023, doi:10.3390/cancers15092651_

Round 1
Reviewer 1 Report
Page 9, line 184-185: “For N1 disease, nodal dissection is performed with further…”
This is true for HPV negative OPC but is not completely correct for HPV positive OPC. The latter may quite often be treated successfully with primary radiochemotherapy. So, I think that this paragraph needs some rephrasing.
Page 9, sub-section “8.2 Response assessment”. Here I miss a reference to Mehanna et al, N Engl J Med 2016.
Additionally, I miss something about PET and radiotherapy with dose-painting.
Reviewer 2 Report
This article reviews the role of positron emission tomography/computed tomography (PET/CT) in the diagnosis and management of oropharyngeal cancers.
Staging of oropharyngeal cancers varies depending upon the presence or absence of human papillomavirus (HPV) directed pathogenesis
The incidence of HPV associated oropharyngeal cancer (HPV+ OPSCC) is expected to continue to rise over the coming decades
PET/CT is a useful modality for diagnosis, staging and follow up of patients with oropharyngeal cancers undergoing treatment and surveillance.
Together, PET/CT enables assessment of primary tumor location, malignant lymph node transformation, and possible distant metastasis or second primary.
recent and upcoming advancements in artificial intelligence and machine learning techniques to improve the utility and efficiency of this modality.
